Enhancing behavior classification of children in dynamic interaction scenes through improved DCNN model

Hao Kexian haokexian@xjy.edu.cn
Xi’an Traffic Engineering Institute , Xi’an, Shaanxi , China
Asif Muhammad
Electronic publication date: 2024 Oct 2
Publication date: 2024
Volume: 10
Electronic Location ID: e2368
Received 2024 May 28; Accepted 2024 Sep 7
Copyright: © 2024 Hao
Copyright year: 2024
Copyright holder: Hao
License: This is an open access article distributed under the terms of the Creative Commons Attribution License, which permits unrestricted use, distribution, reproduction and adaptation in any medium and for any purpose provided that it is properly attributed. For attribution, the original author(s), title, publication source (PeerJ Computer Science) and either DOI or URL of the article must be cited.
License URL: https://creativecommons.org/licenses/by/4.0/

Keywords: Environment creation, Behavior classification, Chinese cultural innovation

Funding: 2023 Shaanxi Teacher Education Reform and Teacher Development Research Project Strategy Research on Kindergarten Education Talent Quality Improvement from the Perspective of Rural Revitalization Strategy SJS2023YB091 2023 Shaanxi Philosophy and Social Science Research Special Youth Project Shaanxi Red Cultural Reflection of Aesthetic Education 2023QN0351 This research is funded by the 2023 Shaanxi Teacher Education Reform and teacher development research project “Strategy research on Kindergarten education talent quality Improvement from the perspective of Rural revitalization strategy (No: SJS2023YB091)” and 2023 Shaanxi Philosophy and Social Science Research Special Youth Project “Shaanxi Red Cultural Reflection of aesthetic Education value in the creation of kindergarten environment (No: 2023QN0351)”. The funders had no role in study design, data collection and analysis, decision to publish, or preparation of the manuscript.

==============================
The rapid development of society makes people pay more attention to the quality of the environment for children’s growth. However, due to the differences of young children, different environments are often needed for cultivation in dynamic interaction scenarios. Therefore, the authors propose an environment creation method for children’s behavior classification to improve the quality of children’s growth environment. Taking the video data of children for a period of time as input, the encoder and decoder are designed to classify children’s behavior and obtain behavior characteristics. After the input image is processed by the backbone network DCNN, two outputs are obtained, which are four times of shallow features and 16 times of high-level features. Aiming at the semantic gap between environmental features and children’s behavior features, the DenseNet model is used to remove the semantic difference between children’s behavior features and environmental features, and the similarity between the two features is fitted as much as possible. The dense blocks obtained by different expansion factors of the network are used for feature connection, so that the model is suitable for feature similarity calculation of different modes. The experimental results show that this method can accurately classify children’s behavior, and the F value is more than 70%, which can provide prerequisites for children’s environment creation. This environment creation model can clearly point out the suitable environment for children and provide a guarantee for children’s growth.

Introduction

With heightened societal and economic sophistication, there has been a growing emphasis on early childhood education. The milieu in which children find themselves plays a pivotal role in their physical and psychological growth. The innovative design of children’s interactive environments is therefore crucial. During the creation process, it is essential to classify children’s behaviors to tailor environments that foster their holistic development (Lee et al., 2016; Mohidin, Ismail & Ramli, 2015). Drawing on the innovative ethos of Chinese culture, this study explores the design of environments tailored to children’s behavior in dynamic interactive settings.

Designing environments to classify children’s behaviors requires consideration of numerous factors. Firstly, one must account for the developmental characteristics of children. Different age groups necessitate varying activities, types of toys, and seat heights (Valente et al., 2021). Safety for young children is paramount; thus, innovative designs should incorporate safety features to prevent falls and injuries, using non-toxic, environmentally friendly, and easy-to-clean materials. Moreover, to nurture children’s innovative capabilities, environmental designs should stimulate curiosity and a desire for exploration (Hammad, Ebaid & Al-Hyari, 2014). Incorporating diverse interactive elements such as operable mechanical equipment and musical toys can achieve this goal.

To accommodate the diverse behaviors of children, environments should cater to a range of needs including play, study, and relaxation. This can be achieved through distinct thematic areas like toy zones, reading corners, and theatrical spaces. Furthermore, flexibility in design, with adjustable seating and tables, ensures suitability for children of varying ages and interests. Sustainability is also crucial; thus, materials and energy use should prioritize efficiency and eco-friendliness (Black et al., 2012).

Chinese culture emphasizes harmony and the unity of the individual with the collective, principles that should guide the creation of children’s environments (Wu, 1985). Architectural designs can draw inspiration from traditional styles such as antique and garden architecture to imbue environments with a Chinese cultural ambiance. Introducing plants symbolizing good fortune in Chinese culture, such as plum blossom, chrysanthemum, and bamboo, enhances children’s understanding of Chinese traditions while appreciating nature (Chen et al., 2018). Cultural activities like tea tasting and paper cutting can be integrated into children’s programs, with dedicated spaces like cultural teahouses or courtyards fostering these experiences. The arrangement of objects within the environment can also reflect Chinese culture, incorporating classic texts like Mencius, the Analogies of Confucius, and the Three-Character Classic to deepen children’s cultural insights through reading. Activities such as traditional Chinese painting, pottery, and games like skipping rope and clappers further immerse children in Chinese cultural practices (Boisot & Child, 1999). By integrating modern technology, a unique environmental design scheme can be crafted that allows children to experience the charm of Chinese culture firsthand.

To fully leverage the role of Chinese culture in environmental creation, we embark on a research project that focuses on behavior-based environmental design for young children within dynamic interactive scenarios, adopting the perspective of Chinese cultural innovation. We integrate traditional Chinese cultural elements into the environmental design, coupling them with modern technological means to create a space filled with interactive and learning opportunities. Through this research, we also give consideration to material selection and energy utilization as auxiliary factors. Our aspiration is not only to enhance the aesthetic appeal and functionality of kindergarten environments but also to subtly strengthen children’s cognizance and interest in Chinese culture, thereby achieving a comprehensive improvement and innovation in their surroundings. The contributions of this study include: Proposing a classification algorithm for children’s behavior to recognize and monitor real-time behavior status.

Introducing a matching mechanism between behavior and environment that adjusts environmental elements based on children’s behavior to promote their physical and mental well-being.

Related works

In 1816, Robert E. Owen established the first nursery school for children under five. Later, Friedrich Froebel of Germany founded the world’s first kindergarten and is revered as its “father”. Froebel emphasized the vitality of children in education and advocated nurturing their creativity within established guidelines to allow their natural inclinations to flourish. Maria Montessori, an Italian educator, is credited as a “kindergarten reformer” for adapting and enhancing previous educational concepts, influencing many kindergartens (Wells et al., 2007; Seshadri et al., 2023; Beltzung et al., 2023; Day & Midbjer, 2007). Japan prioritizes studying children’s behavior and psychology, conducting extensive analyses of kindergarten spaces and advocating for improved environments. German architect Mark Dudek promotes integrating education and architecture to create child-centered developmental environments. Europe’s early childhood education system emphasizes merging education with architecture, respecting children’s rights, and employing diverse teaching methods to facilitate learning complex knowledge (Sethy et al., 2023; Spencer & Blades, 2006). Dudek (2007) advocates understanding children’s psychological needs and designing kindergarten interiors accordingly. Okamura, Kanazawa & Yamaguchi (2007) outlines Japanese kindergarten design principles that respect children’s rights and cultivate positive qualities through case analyses.

Culture refers to the values and value systems shared by a group or society, along with the material manifestations that embody these values. Traditional cultural elements represent one such material entity. Environmental design in art and culture reflects both a material attitude and a way of life, influencing people’s psychology and physiology with its historical continuity and future implications (Jing, 2003; Tang, 1995; Lin, 2017). China’s modern environmental art has evolved through the lens of Western modernization, demanding a deeper understanding of cultural studies. Culture, as a nation’s spiritual essence, boasts enduring historical significance and vitality. The assimilation of Chinese culture with foreign influences represents the essence of Chinese cultural evolution. Designers often employ traditional cultural elements as symbols and creative tools to convey cultural depth and uphold contemporary and national spirits. This cultural phenomenon significantly impacts environmental design art, exemplified by the architectural splendor of Xiamen University (Kim, Wen & Doh, 2010; Roscoe et al., 2019; Liu, Zeng & Liu, 2023). In the early 20th century, overseas Chinese faced oppression from imperialists and foreigners abroad, leading them to integrate Chinese-style roofs onto Western-style buildings as a means of expressing their emotions. Yet, Yue et al. (2020) argues that blending ancient and modern, Chinese and Western design not only preserves national identity but also proves more cost-effective and practical.

In the research on early childhood behavior and Chinese culture, the two fields have always been conducted independently. Early childhood behavior research mainly focuses on understanding and improving the psychological and behavioral development of young children through scientific methods, while Chinese culture research focuses more on the inheritance and innovation of traditional cultural elements in the modern environment. Particularly, combining the rich connotations of Chinese culture with early childhood behavior research not only helps to more comprehensively understand and guide children’s growth but also better transmits and promotes the values and spirit of traditional Chinese culture in early childhood education environments. By organically integrating the research results of these two fields, it is possible to create a more distinctly Chinese early childhood education environment, allowing children to be subtly influenced by cultural immersion in their daily lives, thereby promoting their overall development.

An environment creation method for children’s behavior classification

The development of children’s environments, a cornerstone of educational construction, is now widely recognized for its importance in aligning with modern educational concepts. Chinese cultural innovation in education is integrating with early childhood education through supportive environmental design. By fostering environments conducive to understanding, children experience traditional aesthetics and gain exposure to cultural knowledge, fostering national pride. The author proposes a Chinese culture-optimized children’s environment program (Fig. 1), integrating cultural projects and equipment across teaching, sports, nature, and art areas.

Figure 1 Environment creation program for infant.

Figure created using Microsoft Visio.

The teaching area offers ample resources for young children to explore poetry, lyrics, and other significant literary works from Chinese culture. The sports area features a playground where children engage in activities like Tai Chi, Wuqinxi, and Cuju to enhance physical fitness. The natural landscape area provides diverse terrain and caves for children to explore, play games, and engage in role-playing activities. It includes plants such as plums, orchids, bamboo, and chrysanthemums, imbued with Chinese cultural symbolism, to foster courage and confidence in children. The art zone serves as a natural haven for creativity, where children freely participate in painting, clay sculpture, weaving, and other creative activities. This environment encourages children to appreciate Chinese culture and art, experience the joy of creation, and develop their aesthetic sense, imagination, and creativity.

Deep convolutional neural network

Based on the above schema for creating children’s environments, the author proposes a model for recognizing children’s behavior to help them find their optimal external environment. The development of a convolutional neural network (CNN) enables us to achieve this goal. To achieve the clear classification of children’s behavior, this study utilizes video data of children captured over a period. An encoder-decoder architecture is designed to classify children’s behavior and extract behavior features. Utilizing a deep convolutional neural network (DCNN), two sets of outputs are generated: shallow features at four times the depth and deep features at 16 times the depth.

In the encoder phase, high-level features undergo processing via Atrous Spatial Pyramid Pooling (ASPP), which employs five distinct convolutional operations to produce five outputs, including one 1 × 1 convolution, three 3 × 3 convolutions with varying rates, and global pooling of the image. In the decoder phase, shallow feature dimensions are adjusted using 1 × 1 convolutions, and they are concatenated with outputs from the encoder phase. The decoded image features are then outputted through a 3 × 3 layer and image upsampling operation, culminating in the extraction of children’s behavior characteristics following data processing to achieve behavior classification. The workflow is depicted in Fig. 2.

Figure 2 Overview of the behavior classification model.

Among them, the convolution operation of DCNN has two ways: wide convolution and narrow convolution, namely wide convolution and narrow convolution, as shown in Fig. 3. Let m be a weight vector and s be the vector represented by the input sequence. m is a convolutional filter, and a one-dimensional convolution operation is performed by taking the dot product of the convolutional filter m with each of the m consecutive subsequences of s to produce another sequence c:

Figure 3 The convolution of DCNN.

(1) cj=mTsj−m+1:j

For narrow convolutions, s > m is required and the returned sequence c∈Rs−m+1 is calculated with j in the range [m, s]. Wide convolution does not have any requirement on s and m, which can return a sequence c∈Rs−m+1 with j in the range [1, m + s−1] and sets all values outside the range of sequence s to 0. Wide convolution computes all weights in the filter along with the whole sequence. This makes sense when m is set to 8 or 10. In addition, the wide convolution operation ensures that applying a filter m to an input sequence s will always produce a valid nonempty result c, independent of m and s. Consider the case m > s, narrow convolutions will not be able to handle, but for wide convolutions, you will still get a valid result.

Environment creation model based on children’s behavior

After obtaining the behavior characteristics and classification information of each child, the authors propose an environment creation model based on children’s behavior.

Aiming to bridge the semantic gap between environmental and children’s behavior features, this algorithm employs the DenseNet model to minimize the semantic disparity and maximize the similarity between these features. The research methodology is illustrated in Fig. 4. DenseNet addresses gradient dispersion issues and effectively mitigates overfitting. The Bottleneck Layers in DenseNet utilize 1 × 1 convolutions to reduce the computational complexity of the feature maps. Furthermore, through the use of transition layers, DenseNet is able to adjust the dimensions and the number of channels of feature maps between different Dense Blocks, thereby reducing both the computational load and memory consumption.

Figure 4 The structure of the environment creation model based on infant behavior.

During the similarity fitting task, the network utilizes its concat layer for feature concatenation, facilitating gradient flow and providing additional training supervision. Dense blocks with varying expansion factors enable feature connectivity across different modes, enhancing the model’s capability for feature similarity calculation. Increased network depth enhances complexity, thereby improving resistance to overfitting. The core deduction formula is:

(2) Xl=Hl([X0,X1,…,Xl−1])

(3) H=BN+Relu+Conv(1×1)+Conv(3×3)

where X0, X and Xl-11 represent the feature map, H represents the nonlinear transformation, and [] represents that all output feature layers are spliced together according to channel combination. In a convolution module or even the whole Dense Block, the feature size is unchanged, which is for convenience when splicing. In DenseNet, each layer is directly connected to all other layers within the same module. This design enables each layer to directly access the feature maps of all preceding layers, thereby enhancing the propagation capability of features within the network. As gradients can be propagated directly to earlier layers through these dense connections, gradients are less prone to vanishing even in very deep networks. This characteristic allows DenseNet to train deeper network structures while maintaining good performance. Furthermore, the transition layers in DenseNet are used to introduce nonlinearity between dense connection modules and reduce the dimensionality of feature maps. This design helps reduce computational complexity and the risk of overfitting.

Experiment and analysis

Dataset and implement details

The author utilizes the HBSC dataset (https://gateway.euro.who.int/en/datasets/hbsc/) for performance evaluation. Experiments are conducted using an i5-12500 CPU and RTX 3,080 GPU, with Ubuntu as the operating system and PyTorch as the deep learning framework. The experiment consists of 200 rounds, employing a batch size of 128 and an initial learning rate set to 0.0001. The Adam optimizer is used with a weight decay term of 1 × 10−4. Evaluation metrics include precision, recall, and F-measure, calculated as follows:

(4) Vp=NTPNTP+NFP

(5) VR=NTPNTP+NFN

(6) F=2Vp⋅VRVp+VR

Results and discussion

The author conducted experiments on the HBSC Dataset to classify children’s behavior in dynamic interactive scenes. Several models were selected for behavior classification, including CNN (Lu et al., 2019), long short-term memory (LSTM) (Yu et al., 2019), CNN-LSTM (Mutegeki & Han, 2020), SwinTransformer (Liu et al., 2022), ViT (Truong et al., 2021), and Deit (Touvron et al., 2021). The comparison results are summarized in Table 1. Overall, the model achieved Precision, Recall, and F-measure scores of 70.64%, 71.34%, and 71.72%, respectively. Compared to CNN and LSTM, the F-measure improved by 8.35% and 6.24%, respectively. Compared to CNN-LSTM, Precision, Recall, and F-measure improved by 4.77%, 4.52%, and 5.05%, respectively. Compared to SwinTransformer and ViT, the method outperformed by 1.37%, 1.89%, and 3.17% in F-measure, respectively. Notably, the method excelled in recall due to its use of both wide and narrow convolutions to capture action features across video frames. Figure 5 illustrates the convergence of this model and others during training, demonstrating faster achievement of model parameter convergence. At the initial stage of training, our model demonstrated a rapid convergence speed, indicating its ability to more efficiently extract key features from the training data and swiftly adjust model parameters to approach the optimal solution. As the training process continued, our model maintained this efficient convergence trend and was able to achieve a satisfactory state of model parameter convergence within a relatively short number of iterations. This result suggests that our approach is capable of more effectively handling the complexity and diversity of infant behavior data, achieving both efficient and accurate classification performance.

Table 1 Compare the method with others.

	Precision	Recall	F value	
CNN	62.45	63.93	63.37	
LSTM	60.68	67.01	65.48	
CNN-LSTM	65.87	66.82	66.67	
SwinTransformer	68.23	69.37	70.35	
Vit	69.45	68.33	69.83	
Deit	68.54	68.82	68.55	
Ours	70.64	71.34	71.72	

Figure 5 The training of the method and other methods.

Additionally, to validate the environmental creation model based on young children’s behavior, the researcher conducted an experiment in a real-life setting. Ten young children were invited to participate in testing the method. Following the layout depicted in Fig. 1, the researcher simulated four functional areas: the sports area, teaching area, nature area, and art area, each featuring a variety of Chinese cultural artifacts. Initially, the researcher observed the children for 1 h, applying a child behavior classification model to categorize their behaviors. Subsequently, the behavioral characteristics and classification outcomes were inputted into the environmental creation model based on children’s behavior to predict each child’s preferred environment. Finally, the children were guided through each area and asked to select their preferred environment. The accuracy rate was determined by comparing these choices with the predicted results.

Among them, Transformer, Clip, and Blip are also utilized as comparative models to assess the performance against the environmental creation model based on children’s behavior, as depicted in Fig. 6. Similarly, the author compared the predicted results of the method with the children’s selections, summarized in Table 2. Here, S denotes the sports area, T represents the teaching area, N signifies the nature area, and A denotes the art area. Circle marks indicate the children’s choices, while triangle marks denote predictions made by the model. The table shows the consistency between this method and the children’s choices, achieving an accuracy of 80%.

Figure 6 Comparison with other methods.

Table 2 Test of the application.

	1	2	3	4	5	6	7	8	9	10	
S	▴^	^				▴^			▴^		
T		▴		▴^				▴			
N			▴^				▴^				
A					▴^			^		▴^	

Conclusion

To aid children in discovering the most conducive developmental environments, the author proposes a method for classifying children’s behavior to create tailored environments. By integrating Chinese cultural elements, this approach establishes foundational environments for children’s growth. The article introduces a deep learning-based algorithm for classifying children’s behavior and subsequently designs an environment creation model that uses behavior as a guiding indicator to identify optimal environments. Experimental results demonstrate that this method effectively facilitates comprehensive environment creation for children, guiding the development of environments conducive to children’s growth. the proposed method excelled in recall due to its use of both wide and narrow convolutions to capture action features across video frames. The CNN-based models risk losing multi-modal details, while LSTM and Transformer models may overlook global features, leading to inaccurate classifications. This model addresses these issues by balancing sensitivity and feature preservation in its design, ensuring both performance and efficiency. In the future, we will endeavor to integrate disciplines such as education, psychology, and cultural studies to develop more comprehensive and culturally rich designs for early childhood environments. Additionally, we will incorporate techniques like attention mechanisms and transfer learning to enhance the algorithm’s ability to recognize complex behavioral patterns.

Supplemental Information

Supplemental Information 1 Code and dataset.

Additional Information and Declarations

Competing Interests

Author Contributions

Data Availability

The authors declare that they have no competing interests.

Kexian Hao conceived and designed the experiments, performed the experiments, analyzed the data, performed the computation work, prepared figures and/or tables, authored or reviewed drafts of the article, and approved the final draft.

The following information was supplied regarding data availability:

The code is available in the Supplemental File.

The Health Behaviour in School-aged Children (HBSC).

dataset is available at the European Health.

Information Gateway: https://gateway.euro.who.int/en/datasets/hbsc.

The data is available at Zenodo: None. (2023). Student Behavior [Data set]. Zenodo. https://doi.org/10.5281/zenodo.8337444.

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
