# Peer review of "Enhancing behavior classification of children in dynamic interaction scenes through improved DCNN model"

_PeerJ Computer Science, doi:10.7717/peerj-cs.2368_

## Round 0.1 · original submission · Major Revisions

Dear authors,
The experts in the field have reviewed your manuscript with interest and they are not satisfied with the current quality of the manuscript. Therefore, they have suggested couple of improvements to be made before we reconsider your article. Please also consider the following suggestions of mine before re-submitting.
The is a need to clearly define the problem and its significance. eg why this study is important and who will be potential beneficiaries
Provide more context about the backbone network DCNN and why it was chosen for this task.
Explain the specific role of the encoder and decoder in the classification process
Clarify how the DenseNet model bridges the semantic gap between environmental features and children's behavior features.
please add include more quantitative data and comparative analysis in the experimental section
Proofreading the text for grammatical errors and awkward phrasing to improve readability is a mandatory requirement for this article.

Reviewer 1 ·

Basic reporting

The author proposes a deep learning-based approach for classifying child behaviors to create tailored environments and designs an environmental creation model that identifies the optimal environment using behavior as a guiding indicator. While this research topic possesses a certain level of novelty, the author can enhance the quality of the article by making revisions in the following aspects:

(1) In the introduction, the author mentions that in exploring the design of environments suitable for child behaviors in dynamic interactive settings, the use of materials and energy should prioritize efficiency and ecological friendliness. However, how does the proposed solution reflect this objective?

(2) It is recommended that the author includes an analysis of the cited literature in the related work section. Specifically, the analysis can cover the advantages, disadvantages, or any inspiration the work provides for the construction of the proposed method.

(3) The related work section aims to present the current research progress in the field to the readers. However, the content in the submitted article is rather limited. The author should add articles published since 2022.

Experimental design

This article utilizes the DenseNet model to minimize semantic differences and maximize the similarity between the output features of the DCNN. The author can enhance the article by adding a description of Figure 4, explaining how DenseNet addresses the issue of gradient vanishing and mitigates overfitting.

Before submitting the revised manuscript, the author needs to conduct a thorough check of the symbols, words, singular and plural forms, tenses, capitalization, and colon usage throughout the text.

In the experimental process, the author should analyze the reasons for the performance improvement of the proposed method, rather than simply listing performance data.

Validity of the findings

It is essential to analyze the limitations of the proposed method or potential future research directions, which can provide readers with valuable research ideas.

Reviewer 2 ·

Basic reporting

No comment

Experimental design

no comment

Validity of the findings

no comment

Reviewer 3 ·

Basic reporting

1. The preamble to model design is too long (Line 114-138), more technical analysis of it;

2. Section 2 Related works do not give a detailed introduction on how the existing researches perform feature extraction (the application of DCNN). Most of the literatures in this part are not only too old, but also lack a summary and review of the existing literatures, which does not correspond to the research goal of this paper;

3. The main feature of DenseNet is the dense connections between layers, where the output of each layer is directly connected to the input of all layers after it. Although this design is helpful for information flow and gradient propagation, it also significantly increases the computational and memory requirements, especially when dealing with high-dimensional environmental features and complex child behavior characteristics, the computational cost may be too high;

4. Due to the large number of parameters, the DenseNet model is prone to overfitting when dealing with relatively small data sets (such as the data in children's behavior research). Although mitigated by regularization and data augmentation techniques, overfitting risk still exists, especially when there is high noise or uncertainty between environmental and behavioral features;

5. It is suggested that authors simplify the structure of DenseNet and reduce unnecessary layers and connections for specific tasks to reduce computational cost and the risk of overfitting;

6. Loss curves during training are provided to analyze the convergence speed and stability of the model. In addition, cross-validation or independent test sets are used to evaluate the performance of the model on new data to verify the generalization ability of the model.

Experimental design

OK

Validity of the findings

OK

Additional comments

OK

---

## Round 0.2 · accepted · Accept

Dear authors, thank you for re-submitting the article, based on in the input from the experts, I am pleased to inform you that your manuscript is now acceptable for publication.

Thank you for your contribution.

Reviewer 1 ·

Basic reporting

No Comments

Experimental design

No Comments

Validity of the findings

No Comments

Additional comments

No Comments

Reviewer 3 ·

Basic reporting

accepted in this form

Experimental design

ok

Validity of the findings

ok

Additional comments

ok